# PRISM III Score Predicts Short-Term Outcome in Children with ARDS on Conventional and High-Frequency Oscillatory Ventilation

**DOI:** 10.3390/children10010014

**Published:** 2022-12-21

**Authors:** Snezana Rsovac, Davor Plavec, Dusan Todorovic, Aleksa Lekovic, Teja Scepanovic, Marija Malinic, Mina Cobeljic, Katarina Milosevic

**Affiliations:** 1Department of Pediatric and Neonatal Intensive Care, University Children’s Hospital, Tirsova 10, 11000 Belgrade, Serbia; 2Faculty of Medicine, University of Belgrade, 11000 Belgrade, Serbia; 3Srebrnjak Children’s Hospital, Srebrnjak 100, 10000 Zagreb, Croatia; 4Medical Faculty Osijek, Josip Juraj Strossmayer University of Osijek, Josipa Huttlera 4, 31000 Osijek, Croatia; 5Institute of Medical Physiology “Richard Burian”, Faculty of Medicine, University of Belgrade, 11000 Belgrade, Serbia; 6Department of Pulmonology, University Children’s Hospital, 11000 Belgrade, Serbia; 7Clinic of Dermatovenereology, University Clinical Centre of Serbia, Deligradska 34, 11000 Beograd, Serbia

**Keywords:** ARDS, PRISM III score, outcome, pediatric, mechanical ventilation

## Abstract

Therapeutic recommendations for pediatric acute respiratory distress syndrome (PARDS) include conventional (CMV) and rescue high-frequency oscillatory mode (HFOV) of mechanical ventilation (MV). The pediatric risk of mortality (PRISM) is a frequently used mortality score for critically ill patients. In search of methods to recognize those patients, we analyzed the PRISM III score as a potential predictor of the short-term outcome in MV subjects with PARDS. A retrospective five-year study of PARDS in children on MV was conducted in the Pediatric ICU. Seventy patients were divided into two groups (age group <1 year and age group 1–7 years). The PRISM III score was used to assess the 28-day outcome and possible development of complications. The most common causes of PARDS were pneumonia and sepsis. Male sex, malnourishment, sepsis, and shock were significant indicators of poor outcome. The PRISM III score values were significantly higher in those who died, as well as in subjects requiring HFOV. The score had a significant prognostic value for short-term mortality. There was no significant difference in outcome based on the comparison of two modes of ventilation. A significantly higher score was noted in subjects who developed sepsis and cardiovascular insufficiency. The PRISM III score is a fair outcome predictor during the 28-day follow-up in MV subjects with PARDS, regardless of the ventilation mode.

## 1. Introduction

Acute respiratory distress syndrome (ARDS) represents a life-threatening clinical syndrome that can be extremely challenging to manage. It is a diffuse, inflammatory lung damage affecting both alveolar epithelial cells and endothelial surfaces [1,2]. With the intent to recognize and diagnose pediatric ARDS (PARDS) early on, in 2015, the Pediatric Acute Lung Injury Consensus Conference (PALICC) has provided the critical care community with the first pediatric-focused definition of this complex medical condition [3]. The Pediatric Respiratory Distress Incidence and Epidemiology (PARDIE) study, published in 2019, gathered data from 27 countries and 145 pediatric intensive care units (PICUs) considering the incidence of PARDS using PALICC criteria. The incidence of PARDS was 3.2% and 6.1% among children on mechanical ventilation (MV) [4].

Treating the underlying cause, providing adequate oxygenation and ventilation, and protecting the lungs from a ventilator-induced lung injury (VILI) represent the cornerstone in management of PARDS [1]. Even though mechanical ventilation is the mainstay of PARDS management, it must be considered that the MV itself can initiate and/or aggravate lung damage. In order to minimize these risks, strategies for lung-protective mechanical ventilation have been created [5]. Both conventional mechanical ventilation (CMV) and high-frequency oscillatory ventilation (HFOV) can be used to ensure optimal lung function.

We have already demonstrated in our recent study that oxygenation index on the third day is a very good prognostic indicator for PARDS demanding MV [2]. Likewise, the Pediatric Risk of Mortality Score (PRISM) and the Pediatric Index of Mortality (PIM) can be used for mortality risk assessment. Additionally, it must be noted that other factors such as body weight (BW) and sepsis predict the survival time in patients with PARDS [6].

Considering the fact that the scoring systems can provide information about the performance among PICUs over time, indicate the course of a disease and are easy to use, the goal of this study was to analyze whether the PRISM III score can be used as a predictor of the short-term outcome in mechanically ventilated pediatric patients with PARDS.

## 2. Materials and Methods

This retrospective, observational study was performed at the Pediatric Intensive Care Unit (PICU) at the University Children’s Hospital in Belgrade, Serbia, from April 2011 to March 2016. The ethics committee considered that a post hoc written consent of parents/guardians was not required because the research was conducted on anonymous patients (Ethical Committee of Medical Faculty, University of Belgrade, No. 61206-5210/4-15). It comprises the same cohort we analyzed in our recent publication [2].

Subjects aged from 1 month to 7 years who were diagnosed with PARDS criteria and were mechanically ventilated (MV) during treatment were included in the study. The exclusion criterion was the use of inhaled nitric oxide therapy. Participants were divided into two categories, based on age: infants (up to 12 months) and children (1–7 years old). In the examined period, there were only a few patients older than 7 years who were treated with HFOV, which was not enough for reliable conclusions, therefore, older patients were not included in this study.

Physiological data recorded on the first 24 h of admission to PICU were assessed for obtaining the PRISM III score. A short-term outcome was defined as survival or death, and was assessed on the 28th day of hospitalization at the Pediatric Intensive Care Unit.

We primarily analyzed a pathologic lung condition in subjects, besides the complications frequently seen at ICU and MV settings: cardiovascular insufficiency, multiorgan system failure (MOSF), renal insufficiency, atelectasis, ventilator-associated pneumonia, tracheotomy, hypoventilation, etc. The definition of sepsis was made according to the criteria of the Surviving Sepsis Campaign 2012 [7]. A comprehensive diagnostic panel was used for all subjects (chest radiography, echocardiography, complete blood analysis, and gas analysis). The subjects were carefully monitored and the well-established criteria for placing patients on MV were used [3]. Besides MV, a medical treatment including sedation, inotropic stimulation and antibiotics was implemented when necessary.

All subjects involved in this study were initially ventilated with protective conventional mechanical ventilation (CMV; limited tidal volumes/plateau pressure, inspiratory plateau pressure maximum 28 cm H_2_O). Based on the follow-up of arterial blood gas analysis (ABGs), oxygenation index (OI) and clinical parameters: deterioration of ABGs to 20% of initial values for PaO_2_, PaCO_2_, SO_2_ and pH, as well as other parameters (chest radiography, decreasing PaO_2_/FiO_2_ ratio), the subjects were converted to a High-Frequency Oscillatory Ventilation or, conversely, those without the abovementioned deterioration stayed on the CMV. The mode of ventilation was noted after the initial hour of admission, as well as on the third day.

We used different types of ventilators for conventional and high frequency oscillatory ventilation (General Electrics, Drager, Maquet, and SensorMedics).

We analyzed data by using the SPSS IBM SPSS Statistics v23 software. A *p*-value less than 0.05, was considered statistically significant. Categorical variables were presented as frequency and ratio (%). Continuous variables were reported as mean ± SD, mean (95% CI) or median (IQR), depending on the normality of the data distribution. Statistical significance between the PRISM III score and multiple variables (outcome, MV mode, complications development) was analyzed by the Mann–Whitney U test. The correlation between the PRISM III score and length of MV was assessed by Spearman’s test. The receiver operating characteristic (ROC) curve was schemed to show the sensitivity and specificity of the PRISM III regarding a short-term outcome, as well as to determine the cut-off values. The impact of the PRISM III score, MV mode, and patient characteristics on the outcome was assessed by univariable logistic regression analysis. We used multivariable analysis for the factors that had *p* values ≤ 0.1 on the univariable one, a Cox proportional hazard model, and Kaplan–Meier analysis were also used for outcome.

## 3. Results

### 3.1. Participants and Descriptive Data

During the five-year observational period, 70 children with ARDS were treated at the PICU of our hospital. The study included 40 boys (57.14%) and 30 girls (42.86%). The average age of all of the subjects was 2.82 ± 2.43 years old. Out of all subjects, there were 37 (52.85%) infants younger than one year of age and 33 (47.15%) children aged 1 to 7. Fatal outcome was noted in 31 subjects (44.3%), while 39 children have survived (55.7%). At some point, all of the subjects required mechanical assistance. The average time spent on MV for all children was 11.79 days, where minimum time spent on MV was 1 day. At the moment of initiation of MV, BW below the 10th percentile of predicted value had 12 (17.14%) subjects, while 58 (82.86%) children had BW above the 10th percentile of predicted value. The most common cause of PARDS in our research was pneumonia with 40%, while in 30% of subjects it was sepsis, aspiration in 11.4%, and concomitant cardiac disease in 7.2% of hospitalized children. (Table 1) The etiologic agent of pneumonia were bacteria (22 patients), fungi (2 patients), and viruses (4 patients). Patient data and clinical characteristics on admission are shown in Table 1.

### 3.2. Main Results

We recognized statistically significant variables in our research. The univariable and/or multivariable analyses of the data from our research indicated that male sex and shock were predictors of early mortality (OR 3.25, 95% CI 1.09–9.69, *p* < 0.05; OR 0.175, 95% CI 0.041–0.758, *p* < 0.05) (Table 2).

However, there was no statistical significance in the short-term outcome between the two age groups (OR 0.91, 95% CI 0.23–3.54, *p* > 0.05). Additionally, the PRISM III score was a statistically significant indicator of a fatal outcome (OR 7.44, 95% CI 2.46–22.47, *p* < 0.001; OR 0.40, 95% CI 0.18–0.62, *p* = 0.001). Based on the results obtained from the Cox regression model, our study suggests that malnourishment, sepsis, and PRISM III score of 13 (>13) as a cut-off value are statistically significant indicators of an unwanted outcome (95% CI 1.25–12.24, *p* < 0.05; 95% CI 1.226–8.205, *p* < 0.05; 95% CI 1.042–14.790 *p* < 0.05, respectively) (Table 3).

The values of the PRISM III score on admission were statistically significantly higher in patients with fatal outcome than in the survivors (15.94 ± 4.53 vs. 10.87 ± 6.05; *p* < 0.05). Children from 1 to 7 years of age had statistically significantly higher score values in comparison to infants (15.82 ± 6.02 vs. 10.7 ± 4.82; *p* < 0.05). The receiver operating characteristic (ROC) curve was used to assess the performance of the score as a predictor of short-term outcome in PARDS. The adequate cut-off value for the PRISM III score was 13. Almost 81 % (80.6%) of children whose score value on admission was less than 13, have survived during the follow-up, while 64.1% of subjects whose score was higher than 13, have died. The PRISM III score is a can predict short-term PICU/s mortality in PARDS (AUC = 0.755, *p* < 0.001) (Figure 1).

Using the score of 13 as a cut-off value, sensitivity and specificity in terms of the 28-day outcome were 64.1% and 80.6%, respectively. The Kaplan–Meier survival analysis, based on the cut-off value of 13 for the PRISM III score, indicated that there was a statistical significance between the score and short-term outcome in children with ARDS (χ2 = 5.51, *p* < 0.05) (Figure 2).

Regarding the mode of ventilation, the PRISM III score was statistically significantly higher in subjects requiring HFOV compared to those on CMV (16.8 ± 5.73 vs. 11.5 ± 5.37, *p* < 0.05). The score values did not significantly correlate with the length of mechanical ventilation in our study (*p* > 0.05). Additionally, our research demonstrates no significant difference in the outcome, comparing both modes of ventilation, CMV, and HFOV (OR 1.82, 95% CI 0.43–7.79, *p* > 0.05).

Analysis of the PRISM III score values and potential complications has shown that subjects who scored higher on admission had statistically significantly higher chances of developing sepsis and cardiovascular insufficiency (Z = −2.13, *p* < 0.05; Z = −2.92, *p* < 0.05, respectively). However, no statistically significant associations between the score values and MOSF, renal insufficiency, atelectasis, and ventilator-induced pneumonia were found (Z = −1.53, *p* > 0.05; Z = −0.53, *p* > 0.05; Z = −1.07, *p* > 0.05; Z = −5.59, *p* > 0.05, respectively).

## 4. Discussion

This retrospective study involved seventy subjects diagnosed with PARDS following the PALICC criteria, who were mechanically ventilated during their treatment in the PICU of a single tertiary center. The study population included slightly more males, and was stratified by age into two groups: infants younger than one year of age and children aged from 1 to 7 years. An older age is recognized as a significant risk factor for death [1,8]. However, the outcome on 28th day in our subjects did not differ between these two age groups. Male sex, on the other hand, was a risk factor for poor outcome. Aragao et al. noted the risk factors for death in PICU, for all critically ill patients, including PARDS. On the contrary, in their research, there was no significant influence of sex on outcome, and mortality risk was higher for children in 0–2 years age group of children [9]. This result can potentially be explained by a relatively small sample size in our research. Moreover, BW above the 10th percentile, besides the absence of shock development, was associated with the survival outcomes in our subjects.

The most common underlying etiology of ARDS in our subjects included pneumonia (40%) and sepsis (30%). We did not examine the influence of the type of pneumonia on the outcome, because bacterial pneumonia was convincingly the most common. Despite a slightly higher proportion of cases with sepsis, these findings are in accordance with those already reported by researchers [10,11]. Moreover, since our results showed that sepsis was a good predictor of death, it might have had an impact on a high mortality rate observed in this study (*n* = 31; 44.28%). It is of note that the mortality rate in children with PARDS on MV varies among studies; it is reported to be around 30% in severe ARDS, but ranges from 20% up to 45% [10,12,13]. The proportion is lower in the studies carried out in developed countries [14,15,16]. Aside from the potential differences in the available resources, healthcare providers, and ICU settings, the overall proportions probably indicate a severe illness perse.

The early prediction of outcome in a disease with high short-term mortality could positively impact the recognition of children in need of urgent therapeutic intervention. The PRISM III score is comprised of 17 physiological parameters (mental status, arterial blood pressure, ABGs, etc.) but does not include the modes, parameters and consequential impact of MV. Having that in mind, it is an obvious indicator of disease severity [17,18,19]. Being able to test it in the settings different from the one in which it was originally developed, will further assess its clinical significance [20]. The assessment of its predictive ability among extremely ill children is of great importance [17]. However, the data on the PRISM score value in terms of PARDS are scarce. ARDS often indicates sepsis and MOSF development, and a simultaneous biotrauma by MV paves the road to death outcome [21]. Therefore, we wanted to assess this score as a potentially meaningful early prognostic parameter. In terms of the 28-day mortality, the PRISM III score values assessed upon admission to our PICU, were significantly higher in subjects who died than in survivors (15.94 ± 4.53 vs. 10.87 ± 6.05). Similar results in children with PARDS on MV were noted in the study by Anton et al. The average score in their survivors was 9.1 ± 4.6 and 17.2 ± 13.5 in those who died [22]. Mirza et al. analyzed the PRISM III value in the PICU of general care settings, including children who did not require invasive ventilatory support. Among 62 of their patients who had a score of 10 to 19, only 6.46% have survived, while those with a score of less than 10, over 85% have survived [14]. Kesici et al., found the PRISM III score to be significantly higher in non-survivors, but factors such as higher proportion of subjects with cardiovascular insufficiency influenced on higher scores in both groups compared to our findings [6]. In our study, the age-related differences in the PRISM III score were statistically significant, as children 1–7 years of age had a higher score compared to infants. However, as already stated, the age-related difference in the main outcome was not registered in our subjects. Additionally, the score was statistically significantly higher in subjects requiring HFOV compared to those on CMV. This observation is expected, as the conversion to HFOV mode is recommended as a rescue therapy in children with the progression of PARDS and further deterioration of physiological parameters [3]. When it comes to discriminatory value of this score in our subjects, the ROC curve analysis indicates a fair suitability (AUC = 0.755). With the score of 13 as a cut-off value, sensitivity and specificity were 64.1% and 80.6%, respectively. We found the discriminatory power to be lower compared to those already reported in the PICU settings. Popli et al. showed good discrimination, Mirza et al. the excellent one, with high sensitivity and specificity, and other research also verified the validity of this score in their PICUs [14,16,20,23]. Again, it should be emphasized that our study considered only severe illness demanding MV, whereby it excluded the various groups of subjects treated in PICU. An early initiation of HFOV in our subjects, however, did not influence mortality compared to the group managed exclusively by CMV. These findings confirm the PALICC rationale [3,24]. Furthermore, the score values did not significantly correlate with the length of MV in our research.

Evaluating the association between the PRISM III and the development of complications could also be important in the context of MV-induced biotrauma [24,25]. A significant association was noted among higher score values in subjects with sepsis. This corroborates the influence of sepsis on outcomes and higher mortality rate in our patients. The same holds true for the relation between the PRISM III and the later onset of cardiovascular insufficiency in our study (*p* < 0.01). These results are consistent with those of Avula et al. [26]. Conversely, there was no association between this score and the development of renal insufficiency (*p* > 0.05), as well as MOSF (*p* > 0.05). The studies are indicating an association between the PRISM III score and the development of renal failure and the requirement of renal replacement therapy in the setting of MOSF development [8,27]. MOSF per se has a predictive value for an outcome in these subjects [1,13]. Finally, as we could have expected, no significant influence of the PRISM III score and the onset of atelectasis or ventilator-associated pneumonia was found in our subjects. This stems from calculating this score exclusively on the basis of physiological parameters, without consideration of the MV parameters. Inadequate MV settings could explain the presence of atelectasis.

There are several important limitations of our study, firstly having a small sample size despite a five-year observational period, due to the limited PICU capacity and the omission of subjects without a need for invasive ventilation. Moreover, our research has a retrospective character and relied on the existing medical documentation. We emphasize the importance of further multi-centric studies which could objectively demonstrate the PRISM III score validity in the specific groups of patients.

## 5. Conclusions

The PRISM III score on admission to PICU is a fair predictor of the 28-day outcome in subjects with PARDS managed with MV. We demonstrated its predictive value in both protective CMV and early HFOV, even when there was no significant difference in mortality between these two modes. Besides sepsis and shock development, as well as malnourishment, with an already established importance, the PRISM III score might also be a valuable early predictor of short-term prognosis, favoring adequate initial interventions to those who are most in need.

## Figures and Tables

**Figure 1 children-10-00014-f001:**
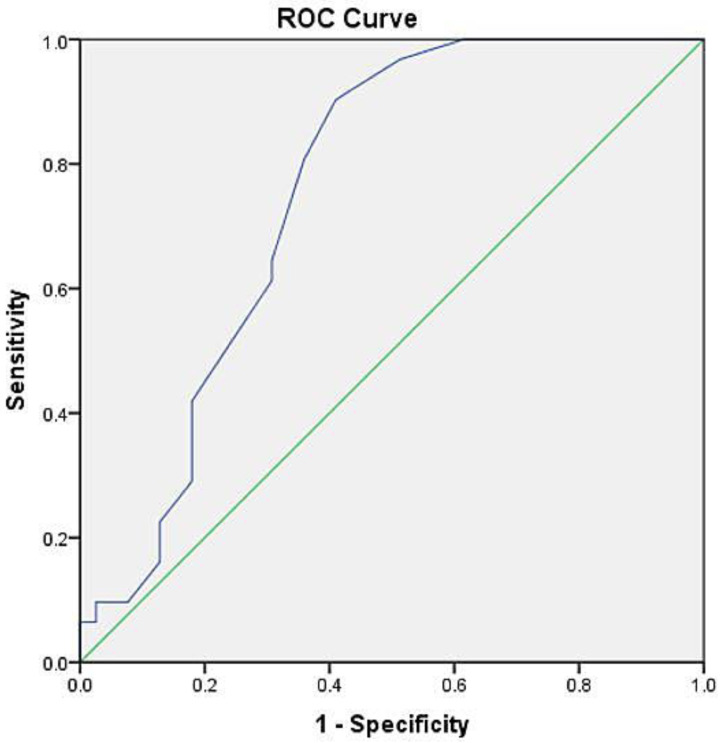
ROC analysis of PRISM III score for prediction on short-term mortality in PARDS. The blue line represents ROC curve for PRISM III score regarding the prediction of the short-term outcome. The green line is a diagonal reference line. Notes: Area under the ROC curve 0.755, SE 0.058, 95% CI 0.641–0.870, *p* < 0.001, Overall Sensitivity 64.1%, Overal Specificity 80.6%, Positive Predictive Value 0.860, Negative Predictive Value 0.641.

**Figure 2 children-10-00014-f002:**
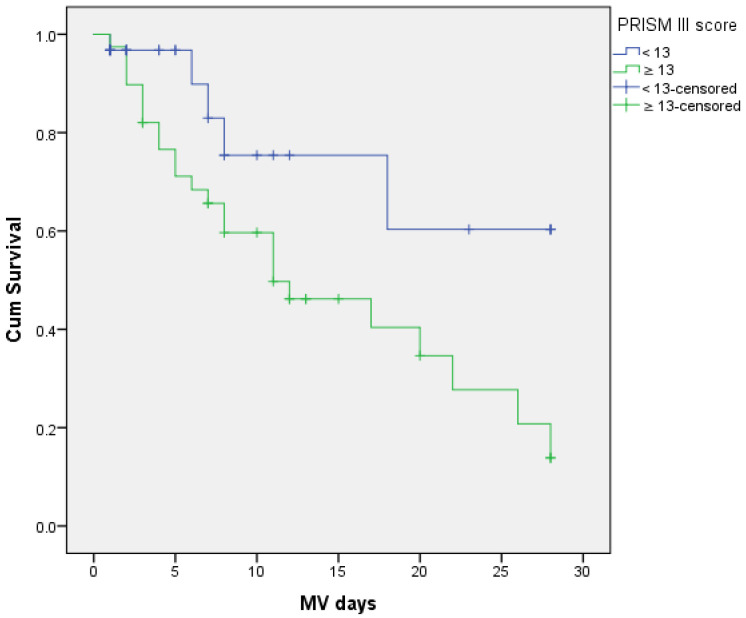
Kaplan–Meier survival curves for PRISM III score values which are < 13 and > 13 for prediction on short-term mortality in PARDS. Notes: Log Rank Mantel-Cox test: χ2 = 4.820; *p*= 0.028, Median: PRISM III < 13 = 28 days (Estimation is limited to the largest survival time if it is censored); PRISM III ≥ 13 = 11 days.

**Table 1 children-10-00014-t001:** Patients Characteristics and Clinical Parameters on Admission and Onset of MV (*n* = 70).

Variable	Frequency	%	
	*n*		
Sex	Male	40	57.1
	Female	30	42.9
Body weight percentile	<P10	12	17.1
	>P10	58	82.9
Blood culture	Positive	25	35.7
	Negative	45	64.3
Hypoxemia at beginning of	<200	38	54.3
MV	<300	18	25.7
Hypoxemia after 1 h of MV	Yes	27	38.6
(20% change)	No	43	61.4
Hypoxemia after 24 h of MV	<200	38	54.3
	<300	9	12.9
Hypoxemia after 72 h of MV	<200	32	45.7
	<300	15	21.4
Mod of MV after 72 h	CMV	49	70.0
	HFOV	21	30.0
pH	<7.3 7.3–7.4 (normal)	48 21	68.6 30.0
	>7.4	1	1.4
PaCO_2_ (mmHg)	<35	5	7.1
	35–45 (normal)	20	28.6
	>45	45	64.3
PaO_2_ (mmHg)	<60	50	71.4
	>60	20	28.6
Variable	Median (25–75%)		
pH	7.21 (7.10–7.33)		
PaCO_2_ (mmHg)	56 (41–67)		
PaO_2_ (mmHg)	52 (45–67)		
Primary etiologies		*n*	%
Pneumonia		28	40.0
Sepsis		21	30.0
Aspiration		8	11.4
Concomitant cardiac disease		5	7.2
Other clinical conditions		8	11.4
Ventilation setting at 0 h + (Mean ± SD; Median (25–75%))	CMV	HFOV	
PEEP (cmH_2_O)	4.47 ± 1.37		
Variable	Frequency	%	
	*n*		
PIP (cmH_2_O)	22.70 ± 3.62		
MAP (cmH_2_O)	11.96 ± 2.42		
FiO_2_	0.84 ± 0.24	0.98 ± 0.06	
Frequency: (CMV–breaths/min; HFOV–Hz)	22 (19–24)	7 (6–7)	
CDP (cmH_2_O)		24.00 ± 1.92	
∆*p* (cmH_2_O)		40.76 ± 3.72	

Notes: Already published in our recent publication [2]; + 0 h equals start of MV. Abbreviations: MV, mechanical ventilation; CMV, conventional mechanical ventilation; HFOV, high-frequency oscillatory ventilation; PaCO_2_, partial pressure of carbon dioxide in arterial sample; PaO_2_, partial pressure of oxygen in arterial sample; TV, tidal volume; PEEP, positive end-expiratory pressure; PIP, peak inspiratory pressure; MAP, mean airway pressure; FiO_2_, inspiratory fraction of oxygen; CDP, continuous distending pressure; ∆*p*, amplitude.

**Table 2 children-10-00014-t002:** Correlation of patient data and mechanical ventilation mode—statistical examination.

Variable	Outcome			Univariable Analysis			Multivariable Analysis		
		Survival (*n* = 39)	Death (*n* = 31)	OR	95% CI	*p* Value	OR	95% CI	*p* Value
Sex	Male Female	27 (67.5%) 12 (40.0%)	13 (32.5%) 18 (60.0%)	2.803	0.863–9.110	0.086	3.255	1.093–9.692	0.034
Age	<1 year	23 (62.2%)	14 (37.8%)	0.912	0.235–3.543	0.894			
	>1 year	16 (48.5%)	17 (51.5%)						
	X ± SD	2.1 ± 2.6	2.4 ± 2.3						
BW percentile	<P10	5 (41.7%)	7 (58.3%)	0.637	0.131–3.093	0.576			
	>P10	34 (58.6%)	24 (41.4%)						
Mod of MV day 1	CMV	30 (61.2%)	19 (38.8%)	1.825	0.428–7.788	0.417			
	HFOV	9 (42.9%)	12 (57.1%)						
Mod of MV day 3	CMV	30 (61.2%)	19 (38.8%)	1.825	0.428–7.788	0.417			
	HFOV	9 (42.9%)	12 (57.1%)						
Shock	Yes No	17 (39.5%) 22 (81.5%)	26 (60.5%) 5 (18.5%)	0.175	0.041–0.758	0.020	0.140	0.044–0.476	0.001
Sepsis	Yes	6 (33.3%)	12 (66.7%)	0.936	0.185–4.734	0.936			
	No	33 (63.5%)	19 (36.5%)						
PRISM III score	<13	25 (80.6%)	6 (19.4%)	7.440	2.463–22.474	<0.001	0.401	0.181–0.621	0.001
	≥13	14 (35.9%)	25 (64.1%)						

Notes: PRISM III score analysis as supplementary to the data we already published [2] Abbreviations: BW, body weight; MV, mechanical ventilation; CMV, conventional mechanical ventilation; HFOV, high-frequency oscillatory ventilation; OR, odds ratio; CI, confidence interval.

**Table 3 children-10-00014-t003:** Indicators for outcome in PARDS on mechanical ventilation (COX regression analysis).

Variable	*p* Value	95% CI	
		Lower	Upper
Sex	0.452	0.579	3.415
Age	0.127	0.738	1.039
BW <10th percentile	0.019	1.253	12.244
Mod of MV 1st day	0.312	0.809	1.945
OI, first day	0.385	0.881	1.205
OI, third day	<0.001	1.226	8.199
Shock	0.054	0.979	14.567
Sepsis	0.017	1.226	8.205
Kidney failure	0.222	0.695	4.810
Tracheotomia	0.168	0.455	92.476
Barotrauma	0.205	0.610	10.053
Partial lung collapse	0.754	0.469	2.847
PRISM III score PRISM III score (cut-off 13)	0.378 0.043	0.845 1.042	1.006 14.790

Notes: PRISM III score analysis as supplementary to the data we already published [2]. BW percentile, third-day oxygenation index, sepsis, and PRISM III score (cut-off value of 13) were significant predictors of short-term survival in children with ARDS treated with mechanical ventilation (multivariable analysis; Cox proportional hazard model). Abbreviations: CI, confidence interval; BW, body weight; MV, mechanical ventilation; OI, oxygenation index.

## Data Availability

The datasets will be publicly available during review or earlier upon reasonable request.

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
