# Peer review of "PRISM III Score Predicts Short-Term Outcome in Children with ARDS on Conventional and High-Frequency Oscillatory Ventilation"

_children, 2022, doi:10.3390/children10010014_

Round 1

Reviewer 1 Report

Thank you for choosing me as reviewer for the study  to determine mortality and outcome in ARDS, which occupies an important place in Pediatric Intensive Care. In general, a well-written study will contribute to the literature by making some adjustments.

1- Why was 7 years old and under taken?

2- Can the cause of pneumonia be divided into viral bacterial?

3- How was the transition standard to HFOV achieved? Is there a standard protocol in the clinic? It can be difficult to provide protocol standard in retrospective studies.

4-Is there a difference in mortality and outcome according to the HFOV device?

5-What is the frequency of obese patients according to body weight percentile and how did the disease progress in obese patients?

Reviewer 2 Report

Gist/Summary:  The authors present a retrospective study in evaluating the PRISM guidelines/scores on folow-up cases with PARDS subjects independent of ventilation mode.   The work is well taken, presented in lieu of Ethics guidelines, but post hoc informed consent was felt not necessary as the work is base don the case/patient history. 

The authors in their backgorund cite Orloff et al. 2019 article which suggests and summarises  current evidence for current standards of care as well as adjunctive therapies.  

The introduction has just 5 references and I suggest the uathors give a gist of more references in lieu of current practices. 

The MV although was abbreviated in a table, it wa snot presented in lins 70-72.  Please do. 

The section 3.2 is called "Main results"  and I suggest authors to give some catchy framework of wonderful results they exploited. For example,  'Statistically significant variables...were recorded..."

The discussions ections in Line s227-233 must be rewritten laying emphasis on Aragau et al. 

There could, however be other constraints like sedation, Adjunctive tehrapies, Nitri Oxide efflux, neuromuscular blockades, surfactants/steroids and prone positions influencing the statistics.  Couldn't authors delve upon them? 

There could be a pictorial methodology inserted

The overall language must be improved 

Scores on a scale of 0-5 with 5 being the best 

Language: 3.5

Novelty: 4

Scope and relevance: 4

Brevity: 3

PS:  Attached small edits in the manuscript
